

# High spatiotemporal resolution pollutant measurements of on-board vehicle emissions using ultra-fast response gas analyzers

Martin Irwin[1], Harry Bradley[1], Matthew Duckhouse[1], Matthew Hammond[1], and Mark S Peckham[1]

[1]Cambustion Ltd., Cambridge, CB18DH, UK.

*Correspondence to:* M.S. Peckham
(msp@cambustion.com)

**Abstract.** Existing ultra-fast response engine exhaust emissions analyzers have been adapted for on-board vehicle use combined with GPS data. We present, for the first time, how high spatiotemporal resolution data products allow transient features associated with internal combustion engines to be examined in detail during on-road driving. Such data is both useful to examine the circumstances leading to high emissions, and reveals the accurate position of urban air quality "hot spots" as deposited

by the candidate vehicle, useful for source attribution and dispersion modelling. The fast response time of the analyzers also enabled accurate time-alignment with the vehicle's Engine Control Unit (ECU) signals enabling correlation with transient air fuel ratio, engine speed, load, and other engine parameters. This helps to explain the causes of the emissions "spikes" which conventional slow response analyzers would smooth out due to mixing within their sampling systems. The data presented is from NO and $NO_X$ analyzers, but other fast analyzers (e.g. Total Hydrocarbons (THC), CO and $CO_2$) can be used similarly.

The high levels of $NO_X$ pollution associated with accelerating on entry ramps to motorways, driving over speed bumps, accelerating away from traffic lights, are explored in detail. The time-aligned ultra-fast analyzers offer unique insight allowing more accurate quantification and better interpretation of engine and driver activity and the associated emissions impact on local air quality.

## 1   Introduction

Urban air quality is of current concern in many of the world's cities (Mayer, 1999; Chan and Yao, 2008), with a particular focus on the health effects of particulate and $NO_X$ emissions (Samet et al., 2000; Kampa and Castanas, 2008), and governments are facing punitive fines for breaching agreed air quality limits (Gov.co.uk, 2016). Internal combustion engines in vehicles contribute significantly to this air quality problem (Kagawa, 2002) and many cities have multiple monitoring stations for mapping their air quality on a low temporal resolution basis.

The measured pollutants at such monitoring stations are affected by the dispersion of the pollutant from its source (e.g. vehicle tailpipes), with both climatic and traffic conditions causing variations in the measured air quality (Khalfan et al., 2017). Some urban authorities issue "live" contour maps superimposed on city street maps to inform the population of the current air quality (Kings College London, 2013) and enable the general population to plan their travel routes accordingly. Techniques





have also been developed to mount mobile gas analyzers on vehicles to map the location of the worst offending streets (Apte et al., 2017).

Ultra-fast response gas analyzers were first developed in 1987 with a time response (i.e. the time taken for the output signal to fall from 90% to 10% of its full scale following a 100% to 0% step input signal to the device, hereafter referred to as $T_{10}^{90}$), of a few milliseconds (Collings and Willey, 1987) and have been used since then for cyclically-resolved analysis of cold start combustion and other very transient engine phenomena (e.g. gear changes, restarting of combustion, and emissions optimisation). Such fast response allows measurement of the transient emissions within a single engine exhaust stroke. The deployment of ultra-fast analyzers combined with accurate Global Positioning Service (GPS; Department Of Defense, 2008) and engine control unit (ECU) data enables the location of tailpipe emissions spikes to be positioned with enhanced spatial accuracy, and the combination of the engine data helps to explain the mechanisms causing the emission of such pollutants.

Urban driving is set by the legislation to constitute a significant portion (no less than 29% by distance) of an RDE emissions test (EU Commission, Article 2016/427, 2016) in recognition of society's reliance on urban transport. Urban driving generally includes numerous transient features. For example: acceleration away from traffic signals, stop/start congested traffic, traffic calming measures such as speed bumps, and awaiting clearance from oncoming traffic to proceed down narrow streets. The accelerations and decelerations intrinsic to negotiating such impedances often have associated spikes of emissions (e.g. a single gear change comprises a series of engine speed and load transients). This contrasts strongly with a typical cruise cycle on uncongested highways, where few transients occur (except perhaps for overtaking) and the tailpipe emissions can be relatively low over a much larger distance (Schmidt, 2017).

It is worth noting that engine emissions are typically very high when the engine is first started until the catalyst-based aftertreatment system becomes active. On gasoline vehicles, this is largely temperature dependant taking approximately 30 seconds in modern vehicles but many hundreds of seconds for the SCR (Selective Catalytic Reduction) $NO_X$ abatement systems in modern diesel engines. During this warm-up time, engine pollutants pass to the environment largely unabated. Thereafter, aftertreatment systems are generally excellent at cleansing engine exhaust pollutants during steady state engine conditions.

The vehicle's ECU data (specifically the exhaust or inlet mass flow, discussed in more detail below) has been used to convert the raw analyzer ppm concentration measurements to a $g/s$ and $g/km$ value, which is a more relevant data product for atmospheric modelling of the pollutant dispersion.

Portable Emissions Measurement Systems (PEMS) have been developed by a number of manufacturers to comply with the latest EU emissions regulations (EU Commission, Article 2016/427, 2016) but their response times are of the order of $1$ s, with a further "delay time" owing to the transit time of sample gas from its source to the analyser. Their response times are further compromised by the additional pipe volumes required to support the exhaust mass flow measurement system such that the resulting response time makes the emissions data difficult to align accurately with ECU and spatially accurate GPS data. Therefore, it is much more challenging to resolve accurately emissions spikes due to the smoothing effect of PEMS.

One of the main challenges of RDE test work is the unavoidable variability in testing conditions. The emissions on a given route can be affected by any of the following list of factors and more; fuel blend, ambient temperature, pressure and humidity, driving harshness, time and dates of travel (prevailing traffic conditions), type of vehicle, timing of gear changes (Crombie





et al., 2016). To solve RDE transient emissions problems, vehicle manufacturers are identifying the type of transient features which are causing emissions issues, replicating the transient within the controlled conditions of a laboratory and then solving the issue, often with the use of fast response emissions analyzers. Such analysis is beyond the scope of this paper.

## 2 Instrumentation

The methodology outlined in this paper is generally-applicable to any vehicle with reasonable access to the exhaust system. This paper contains minimal discussion about the engine and aftertreatment causes of these emissions to keep the focus on the technique and the instrumentation. The case studies discussed herein are reliant on data obtained using a single two-channel analyzer operated in a configuration described below and demonstrates some, but not all, potential features of interest when using such a technique.

Figure 1 is a schematic showing the layout of the various components. A GPS module is fixed to the roof of the car, and connected to the logging computer using USB. GPS data points are subject to the standard errors inherent to GPS Bajaj et al. (2002). These errors include but are not limited to atmospheric water vapour causing propagation delays in the radio signal and multipath errors where the signal reflects off nearby buildings, confusing the receiver. Dilution of precision due to the unfavourable positioning of satellites is also a possible source of error, meaning it can be very difficult to quantify the

inaccuracy of any measurements taken. At any time the accuracy of the GPS measurement could be between $2.5 - 10$ m CEP (Circular Error Probable). However, GPS is a precise measurement technique with finest achievable resolution of $11$ cm at $52°$N. GPS is logged at $10$ Hz, though this was set to $1$ Hz for the London drive cycle due to this being an initial test of the technique. The $100$ Hz emissions data was provided by a two-channel fast-response chemiluminesence analyzer (CLA; Reavell et al., 1997), situated in the rear of the vehicle cabin, powered via an inverter from a large capacity $12$ V battery so

as to avoid unnecessarily loading the engine. The analyzer data was binned according to GPS midpoint such that on average the mean of $10$ analyzer concentration values are mapped onto a single GPS point, resulting in a high precision spatiotemporal emissions measurement (rather than just one concentration measurement per spatial point). The analyzer was reconfigured from its standard laboratory layout to minimise its size and power consumption, making it suitable for on-board use. Engine data was logged from the vehicle's On-Board Diagnostics (OBD) port, which enables access to the controller area network

(CAN) data from the ECU. This data is available at a maximum rate of $10$ Hz.

The $1100$ mAh, $12$ V battery and $1000$ W inverter supplied power to the gas analyzer for >$120$ mins. Not shown in the schematic are two video cameras with audio recording: one aimed at the gear selector and another aimed forwards through the front windscreen. The video cameras record to internal SD cards which are later time-aligned to the fast on-board measurement data.

The analyzers were tested for vibration signal insensitivity by logging the output to calibration gas while subjecting the rear of the vehicle to shock vibrations whilst stationary. All data is logged to one computer running a custom data acquisition program written in LabVIEW 2015. The data acquisition system merges the analogue, CAN and GPS data on a common time-




base for ease of processing (both logged at 10 Hz). The program also outputs a spatially resolved dataset where each data point corresponds to a physical location with all faster data averaged between each time step.

Regarding sampling location, the gasoline emissions data recorded for this study was taken from an emissions development vehicle which was being used to establish direct links between engine operating parameters and tailpipe emissions; a sampling

pipe being fitted through the vehicle floor to a sampling point pre-muffler but post 3-way catalyst (i.e. all data is post-catalyst). This yielded excellent time alignment for engine comparison purposes. The diesel data was taken from a vehicle for which floor drilling was not favoured and therefore the tailpipe data was taken post-muffler. The effect of the diesel vehicle's muffler on the measured ppm signal yielded a delay of approximately 0.4 seconds equivalent to an artificially early appearance of emissions of about 3 metres distance in the pollutant deposition positioning at 30 km/h vehicle speed. This could be corrected

for by post-processing if desired. For air quality interpretation alone, tailpipe measurements are adequate, but pre-catalyst measurements are required to correlate engine & driver inputs more accurately to these emissions.

## 2.1 Diesel-fuelled vehicle

Diesel emissions were measured from a Euro 5 compliant, 2.0 litre, diesel passenger car with 96,000 kilometers of operation, fuelled with standard UK pump diesel fuel (European Standard EN 590:2013, 2013). The vehicle was in good repair and in

current use. For fleet context, Diesel vehicles classified Euro 5 and earlier account for approximately 87% of the UK's licenced Diesel cars in 2016 (DVLA, 2017). One channel of the analyzer was configured for measurement of [NO] and the other for measurement of [NO]+[$NO_2$]=[$NO_X$] via an $NO_2$ converter (which decomposes $NO_2$ to NO). The sampling pipes from these two channels were connected to a single stainless-steel sample pipe entering the vehicle's tailpipe, penetrating 200 mm inside the rear muffler. The resulting $T_{10}^{90}$ response time of this sampling arrangement was approximately 20 ms, but engine exhaust

transients are heavily damped by mixing in the muffler volume. However, depending on exhaust flow rate, transients can be seen with an observed $T_{10}^{90}$ rise time of approximately 100 ms, suggesting that an installation of this manner is suitable for this application. This sampling technique measures the deposition concentrations at street level (e.g. before turbulent dilution in the vehicle's wake). The emissions data was logged at 100 Hz.

The London route was chosen because it has been used by Transport for London for emissions studies in the past (unpub-

lished), and the drive duration was 2.5 hours for the 53 km of this route. An urban route around Cambridge UK passing near continuous air quality monitoring stations was also designed and used for comparison of diesel and gasoline vehicle emissions.

## 2.2 Gasoline-fuelled vehicle

Gasoline emissions were measured with a Euro 4 compliant, 1.6 litre, turbocharged GDI passenger car, with 80,000 kilometers of operation, fuelled with standard UK pump unleaded gasoline fuel (European Standard EN 590:2013, 2013). The vehicle was

in good repair and in current use. For fleet context, gasoline vehicles classified Euro 4 and earlier account for approximately 66% of the UK's licenced gasoline cars in 2016 (DVLA, 2017).

The sampling points on the gasoline vehicle were different from the diesel vehicle as it was anticipated that the $NO_X$ emissions would be relatively low ($NO_X$ being mainly a byproduct of diesel-powered internal combustion engines) (Heywood,





1988). The first channel of the fast CLA was fitted upstream of the three-way catalyst in the exhaust (hereafter referred to

as "engine-out") and the second channel was fitted downstream of the three-way catalyst (upstream of the muffler, hereafter
referred to as "tailpipe").

The gasoline vehicle was driven around the Cambridge route and by comparing the engine-out to the tailpipe data, the
5   conversion efficiency of the three-way catalyst could be calculated.

## 3   Results

Continuously logged data (two-channel gas measurement, OBD data, and GPS location) when combined with exhaust mass
flow (see below) results in two main data products: 1) gaseous mass emissions (e.g. $NO_X$ in g/s), with vehicle speed and
other diagnostic data, and 2) spatially-binned exhaust mass emissions in g/km. Data product (1) is essentially gas analyzer
10   "raw data" converted into mass, and the time resolution is very high, limited only by the analogue data collection of the gas
analyzers (and for vehicle speed, limited by the CAN-bus). Data product (2) is limited by the GPS acquisition speed (e.g.
Hz), and the emissions data are "binned" according to GPS midpoints. In binning the data, the mean values between each
GPS bin midpoint are calculated, resulting in an average mass emission across the GPS point.

Mass Air Flow (MAF) data has been used when available from the ECU (i.e. the Diesel Euro 5 vehicle), and where MAF
data is not available from the ECU it has been calculated using intake air temperature, engine speed, manifold pressure, the
dimensions of the engine, air/fuel ratio, and an estimation of volumetric efficiency based on empirical calculations (i.e. the
gasoline Euro 4 vehicle).

Since September 2017, EU vehicle emissions legislation requires new vehicles to comply with Real Driving Emissions
(RDE) requirements (EU Commission, Article 2016/427, 2016) in an attempt to make vehicles less polluting over an extended
(and largely unpredictable) set of operating conditions compared with the standard drive cycles which have been used to date
(e.g. NEDC; United Nations, 2013). One of the main challenges in capturing the more dynamic conditions of RDE is the
harsh transient driving behaviour for which fast response emissions analyzers are well suited. For the purposes of illustrating
the applicability of this technique, the results have been broken up into several representative aspects of real world driving,
highlighting areas of interest shown in Figure 2 that require fast measurement to capture transient phenomena: 1) traffic lights,
2) motorway ramps, and 3) speed bumps.

### 3.1   Traffic lights

Figure 3 shows the Diesel $NO_X$ emissions increasing from a 20 mg/s baseline when stationary, associated with sustained
lean operation of the engine during the idle period, to around 70 mg/s during the accelerative phases (increased engine load)
following each gear change after pulling away from the traffic lights. The $NO_X$ emissions increase slightly after vehicle speed
increases (i.e. accelerates) at around 3266 s due to gaseous mixing and gas transit time in the exhaust of the vehicle. This would
be avoided by sampling upstream of the muffler where sampling port installation, not desirable on all vehicles, is required. The
right side of Figure 3 is a satellite view of central London, with $NO_X$ emissions in g/km shown as a function of colour of the





mean value binned per GPS mid-point, for the route driven (driving direction indicated). The section of the drive shown on the left is contained within the red box.

## 3.2 Motorway ramps

Figure 4 shows the data collected on the exit and entry slip road to a 70 mph ($\sim 112$ km/h) dual-carriageway. On the left, a time series plot shows NO emissions from the EURO 4 gasoline engine alongside vehicle speed. Phase 1 of the manoeuvre shows the deceleration off the dual-carriageway on approach to the first roundabout. As expected when slowing down, load on the engine is very low, and emissions are therefore minimal. Phase 2 shows the navigation of the first roundabout followed by an acceleration and gear-change between the two roundabouts. Immediately after the gear change, a very short duration spike of NO can be seen at 923 s. The magnitude of this spike is in excess of 120 mg/s - a considerable emissions peak. Using much slower conventional PEMS equipment ($T_{10}^{90} \sim 1$ s), this highly time-resolved event would be significantly delayed, and smoothed out over a longer period making its location difficult to place. The inclusion of simultaneous GPS data identifies such emissions hot spots spatially. The scale of the spatial markers on Figure 4 correlate colour to the mean NO emissions at that time and location (i.e. values over 50 mg/km are coloured the same as at 50 mg/km). A black data point can be seen on the exit of the first roundabout to show the emissions at 923 s. A further spike in emissions was observed at the second roundabout due to a second deceleration, followed by an acceleration. Phase 3 shows NO emissions spikes correlating with gear changes and high load acceleration as the vehicle joins the main dual-carriageway. The increased NO emissions with each high load acceleration following each gear change are easily visualised on the GPS map plot on the right.

## 3.3 Speed bumps

The data from the Diesel vehicle most clearly shows the location of speed bumps, due to the significantly higher $NO_X$ emissions associated with all accelerative phases (i.e. higher signal compared with gasoline measurements for the same test).

Figure 5 shows $NO_X$ and NO emissions and vehicle speed against time, with a spatial plot of $NO_X$ emissions in mg/km on the right. The $NO_X$ and NO emissions vary with vehicle speed over three subsequent speed bumps, labelled numerically on both the time-series and spatial plots. The nature of speed bumps is to force the driver to slow significantly before accelerating back up to cruising speed, and are often located immediately outside schools or in residential areas as a safety measure. The largest emissions are again associated with the acceleration which occurs immediately following each speed bump, as the driver tends to accelerate towards the speed limit. The acceleration is briefly interrupted by a gear change which is easily identified by the significant reduction in $NO_X$ emissions followed by an immediate sharp increase. The fast response time of this setup illustrates the high temporal resolution of each of these fast transient features. Figure 6 shows the characteristics associated with a single speed bump. Where decelerations are fairly sharp and fuel shut-off occurs, a sharp drop in $NO_X$ is observed as there is no combustion producing $NO_X$ emissions. Further, the drop in emissions at 2218 s is due to a gear change; illustrating that acceleration (gradient of the velocity profile) alone is not a suitable proxy for emissions. The inclusion of traffic calming measures such as speed bumps outside schools may reduce average vehicle road speeds, but appears to increase local pollution significantly.



## 4 Discussion and Conclusions

Ultra-fast response engine exhaust emissions analyzers have been adapted for on-board vehicle use, and when combined with OBD and GPS data allow, for the first time, numerous transient features associated with the on-road driving of internal combustion engines to be examined in detail. On-board, the analyzer's sampling rate of 100 Hz captures emissions transients

that would otherwise be lost or smeared when using conventional PEMS equipment or other slower analyzers.

The analyzers were adapted for on-board use by reducing their size and power requirements far below the original laboratory specification thereby allowing for at least a two hour operating interval. Further, OBD and GPS data were logged simultaneously with exhaust emissions such that the aforementioned transient features can be analysed with a high temporal resolution and with precise location. Using exhaust mass flow (or a derived value), $NO_X$ concentrations were converted to $g/s$, and then

binned into the simultaneously logged GPS points as $g/km$, resulting in a spatiotemporal data product that can be overlaid onto satellite mapping services (e.g. Google Maps, OpenStreetMap) for the easy identification of emissions hot-spots. A selection of hot-spots was explored in the data analysis, giving insight into the transient features associated with the high emissions at these locations. Traffic lights, motorway ramps, and speed bumps were three examples of daily driving conditions that benefit from fast-gas measurement in terms of identifying under what engine conditions spikes in emissions occur, and the resultant

emissions hot spots in terms of geographic location. Namely, the accelerations (including their gear changes) caused by traffic conditions and road layout are the cause of these high emissions. $NO_X$ was chosen for this study as being one of the main urban air quality pollutants of concern, but fast response THC, CO and $CO_2$ analyzers can also be used in a similar manner.

Future work will concentrate on the fast processing and presentation of such recorded data and also expanding the number of vehicle types tested, in addition to more accurate GPS including vehicle speed.

*Acknowledgements.* Transport for London (TfL) for supplying a central and west London candidate route.



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



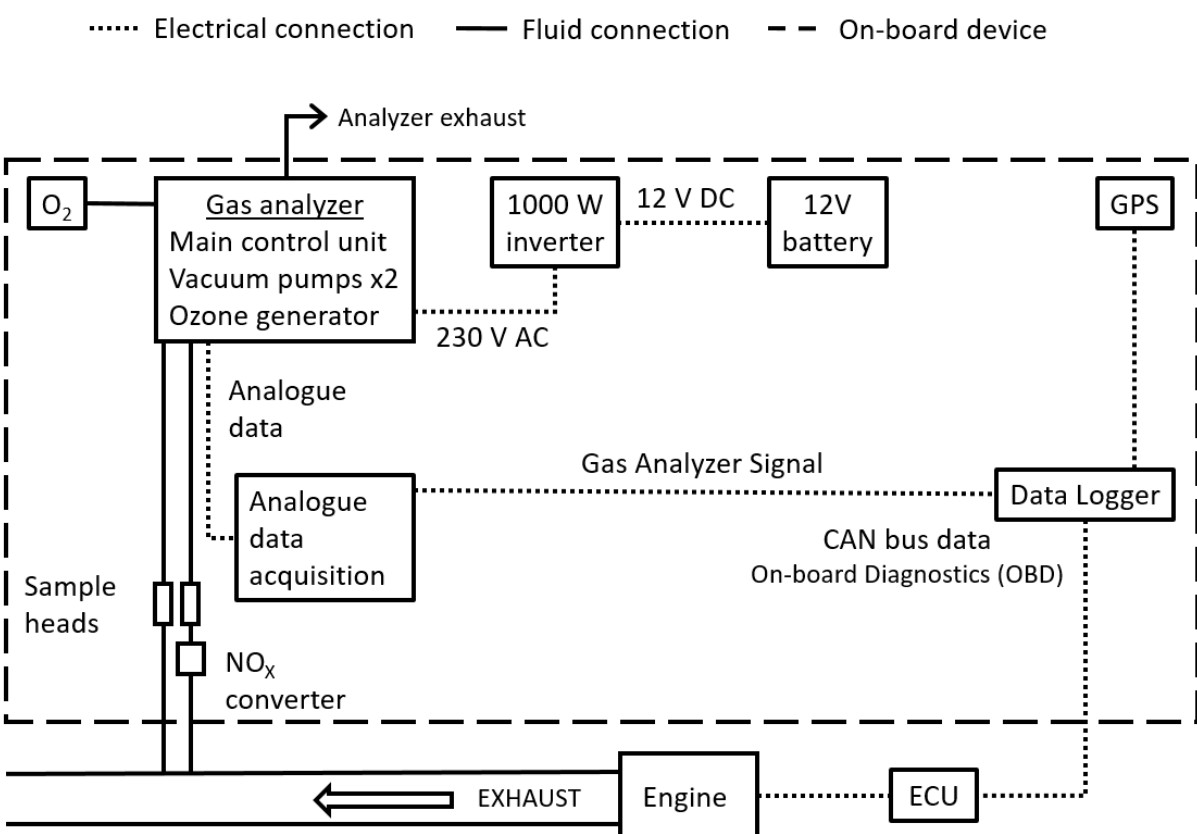

**Figure 1.** Schematic showing the layout of the various components required for the on-board fast gas analyzer measurements. Dashed lines represent electrical connections, and fluid connections are represented by solid lines.




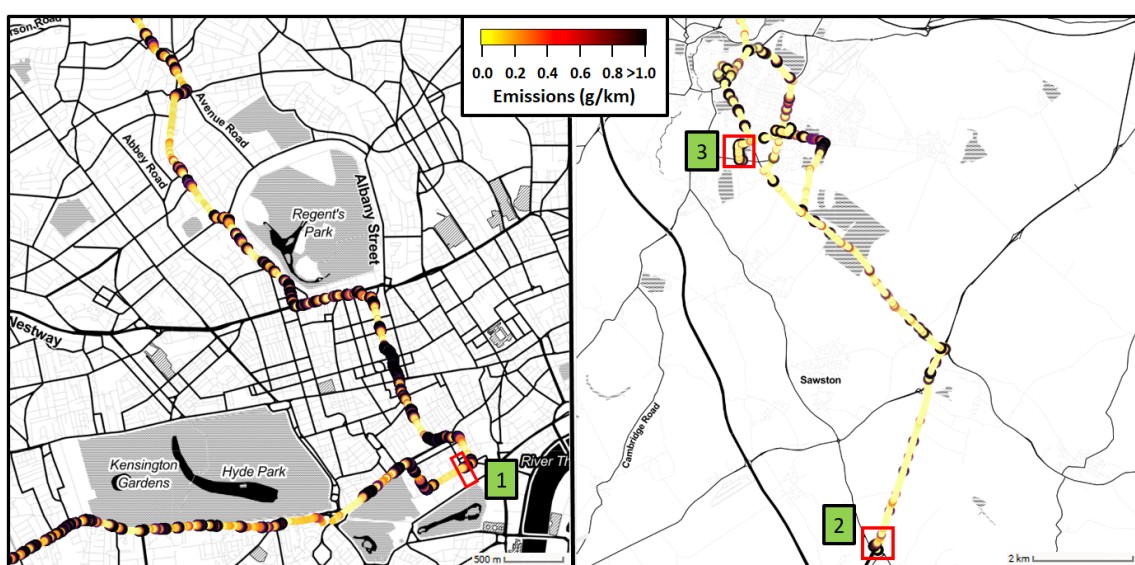

**Figure 2.** Overview maps showing $NO_X$ emissions from the Diesel (left) and gasoline (right) vehicles driven around London and Cambridge (left and right, respectively). Tailpipe emissions higher than 1 g/km per GPS point are circles coloured in black. Points of interest are numbered: 1) traffic light, 2) motorway ramp, 3) speed bumps.





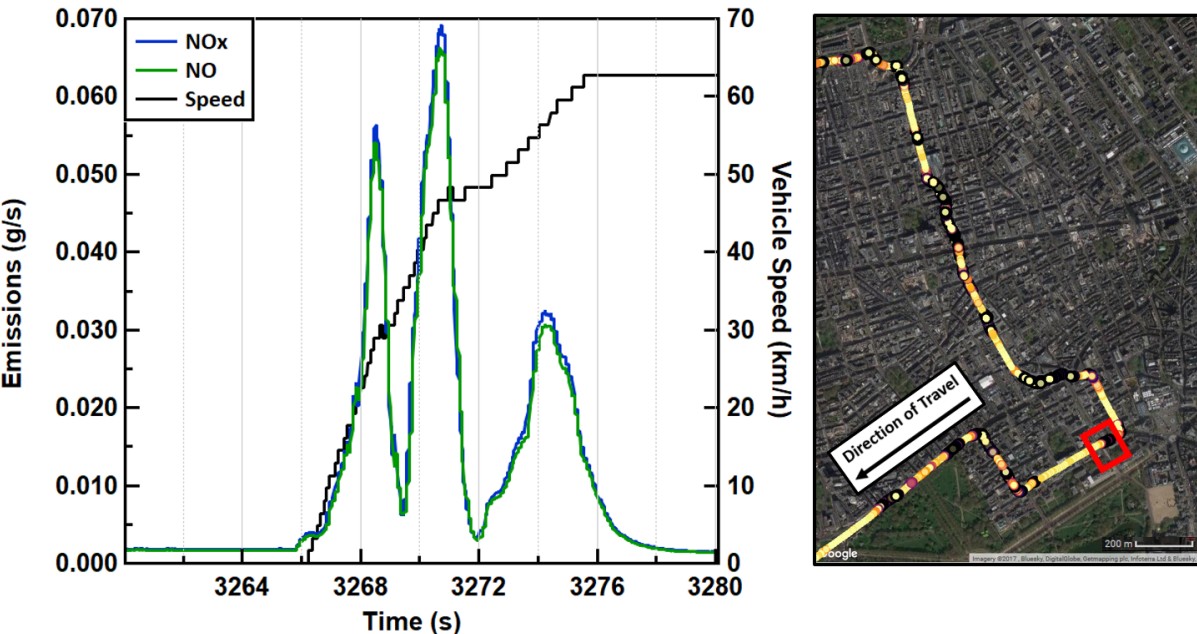

**Figure 3.** Time series of $NO_X$, NO, emissions in g/s with vehicle speed of a traffic light pull-away in central London, UK. The red box represents the spatial component of the graph data, shown in the context of the city area.

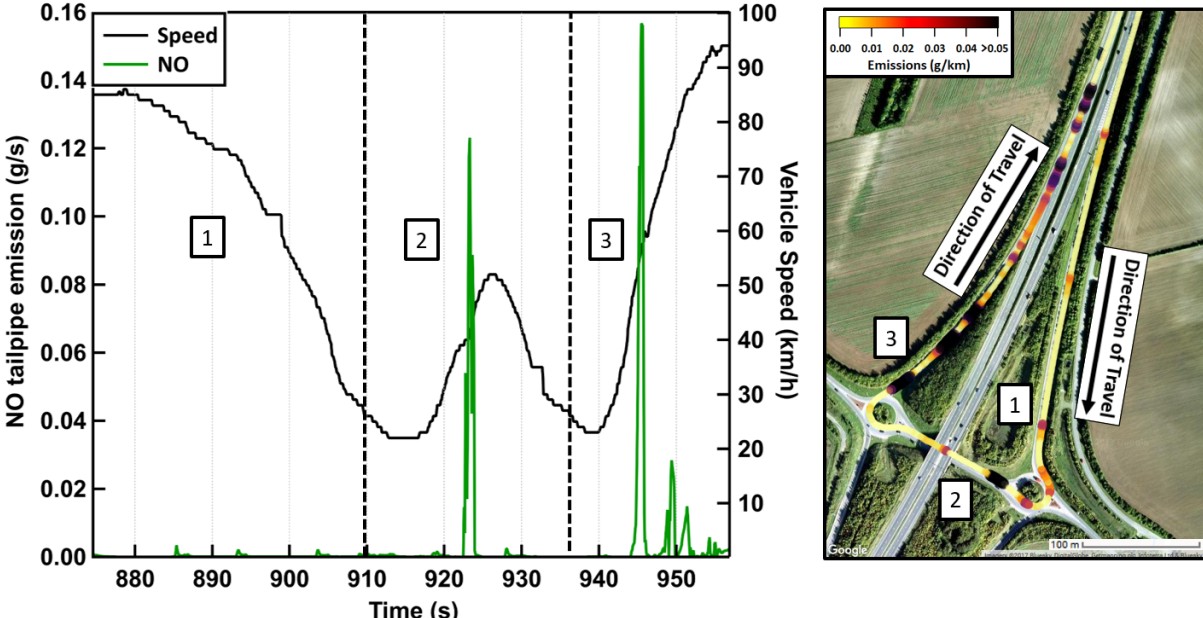

**Figure 4.** Emissions measurements associated with coming off, traversing, and rejoining a motorway shown in three sections; 1. exiting the motorway, 2. crossing under the motorway, and 3. rejoining the motorway. The graph on the left shows the fast NO emissions (g/s) with vehicle speed on the right axis, and the plot on the right shows the route of the vehicle, coloured by NO emissions in g/km. Note the different units on the scales for each plot, and that values over 0.05 g/km are all coloured black in order give sufficient dynamic range on the colour scale.



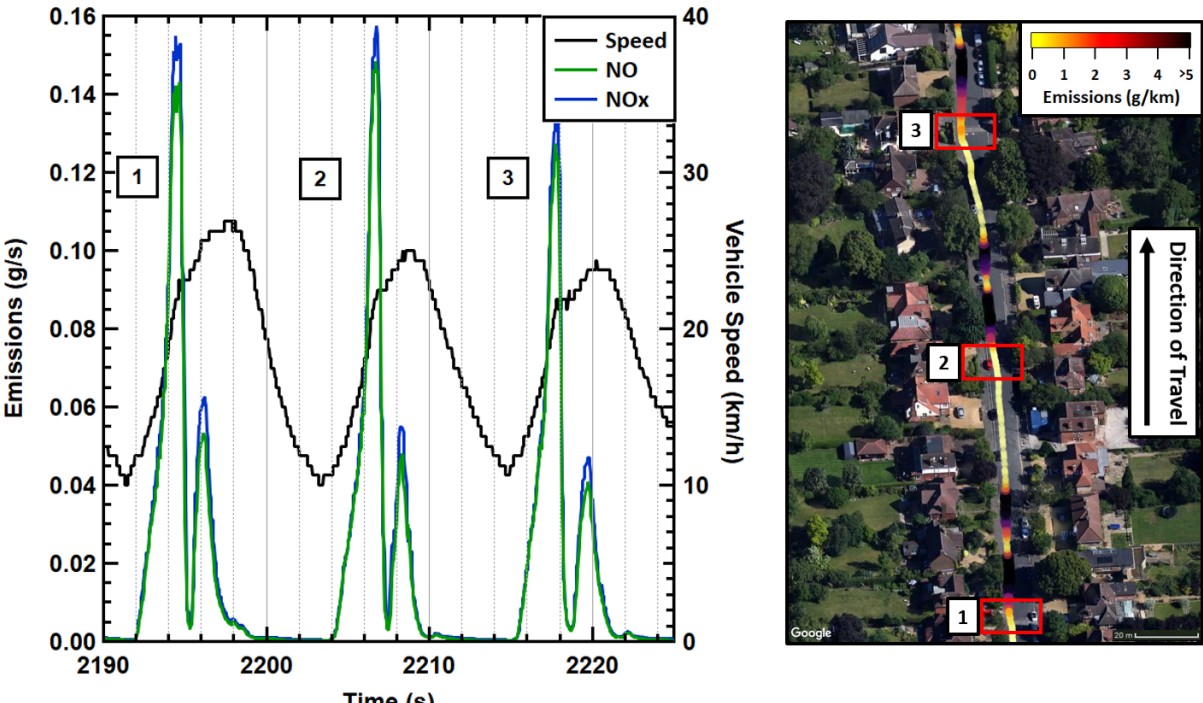

**Figure 5.** NO$_X$ and NO emissions in g/s with vehicle speed, showing the transient behaviour associated with driving over three speed bumps in immediate succession. The plot on the right shows the geolocation of each speed bump (shown in a red box) with a coloured GPS trace showing emissions in g/km.





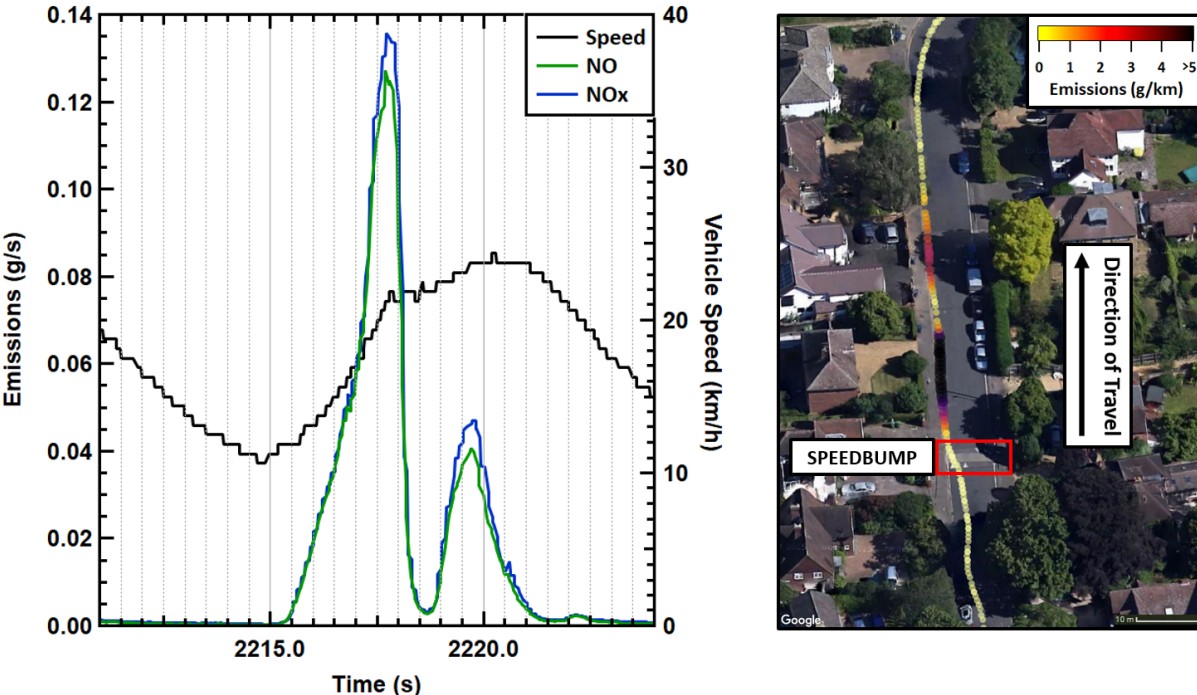

**Figure 6.** $NO_X$ and NO emissions in g/s with vehicle speed, showing the transient behaviour associated with driving over a single speed bump in detail. The geolocation plot on the right shows the emissions in g/km associated with the vehicle's negotiation of the speed bump.