# Peer review of "High spatiotemporal resolution pollutant measurements of on-board vehicle emissions using ultra-fast response gas analyzers"

_Atmospheric Measurement Techniques, 2017_

## Referee Comment (RC1) · Anonymous Referee #1 · 26 Feb 2018

The researchers present an interesting setup in which NOx emissions of cars can be monitored under real driving conditions with a high temporal resolution. Their technique detects short but very strong emission peaks which can't be detected with conventional methods (having a slow and delayed response time). In their driving experiments they detect strong variations of emissions related to acceleration/gear shifting at e.g. speed bumps, traffic lights, motor way access lanes.

I recommend publication after minor changes, although I feel that the paper could gain further scientific relevance with a more in-depth data analysis of the driving experiments. Apparently, both research vehicles comply emission standards when averaged

over time, but exceed emission standards for short time periods. The high spatio-temporal measurements are very relevant for e.g. detecting traffic emission hot spots in an urban areas and assessing possible air quality gain by promoting a steady traffic flow. Although the authors wish to "keep the focus on the technique and the instrumentation" (page 3, line 6), a better data analysis will lead to more interesting conclusions or recommendations (e.g. for reducing traffic emissions by changing driving styles or adapting urban infrastructure), and would give more direction to follow-up research.

Minor comments:

In Section 2 (page 4, line 5) it is explained that the emissions from the gasoline car are sampled "pre-muffler but post 3-way catalyst", which is in conflict with the description in Section 2.2 (page 5, line 1-3) which state that NO is sampled both upstream and downstream of the catalyst. In the latter case: can the authors present some numbers about their measured efficiency of the catalyst? Is this as expected? How does it react under transient driving conditions?

Page 2, line 11: abbreviation RDE is used, while only explained at page 5 (line 19) for the first time.

Page 4, line 13-15: maybe it should be remarked that the diesel vehicle is not equipped with a catalyst.

Page 4, line 32-33: "(. . .) as it was anticipated that the NOx emissions would be relatively low (NOx being mainly a byproduct of diesel-powered internal combustion engines)". I presume that the authors mean NO2 instead of NOx (and that therefore there was no need to use an NO2 converter).

Page 5, line 27: "Figure 3 shows (. . .) increasing from 20 mg/s baseline when stationary". From the figure I would estimate 2 instead of 20 mg/s.

Page 5, line 27-32: The authors are measuring both NO and NOx, so I would expect some words about their findings in the NO/NOx ratio. Is the NO2/NOx ratio approximately stable (also during the acceleration phases)? If so, is it relevant to keep measuring NO2 in future experiments?

Section 3.2, page 6, line 9-11: "Using much slower conventional PEMS equipment (...), this highly time-resolved event would be significantly delayed, and smoothed out over a longer period making its location difficult to place." I would also remark that the duration is so short (∼2s) that its magnitude would be missed by PEMS, having a response time of about 1 second.

Section 3.2, page 6, line 12-13: To put things into perspective, I would include a remark that the current Euro emission standards are 60 mg/km (petrol) and 80 mg/km (diesel).

Section 3.3, page 6, line 19-20: From Figure 2b I learn that also the petrol car has been driven over these speed bumps. Why not include a comparison between the petrol and diesel vehicle emissions regarding speed bumps, as more or less promised on page 4, line 26: "(...) used for comparison of diesel and gasoline vehicle emissions."?

Page 6, line 30: "illustrating that acceleration (...) alone is not a suitable proxy for emissions.". Why not? If I look at Figure 6a, I see two different velocity gradients (roughly 2216.0-2217.5s and 2218.5-2220.0s), corresponding to different emission peaks. If the sampling rate of the speed permits, it would be insightful to overplot this graph with the vehicle's acceleration.

Section 4, page 7, line 4-5: "the analyser's sampling rate of 100 Hz captures emission transients that would be lost or smeared when using conventional PEMS equipment or other slower analysers". I would include this important statement also in the abstract.

Figure 1: Consider to include arrow heads on the dotted lines to indicate direction of data flow.

Figure 3b: Missing legend/color bar. It would also be interesting to see the location of all traffic lights along the driving path.

---

## Referee Comment (RC2) · Anonymous Referee #3 · 11 Apr 2018

This is an interesting paper showing how emissions of nitrogen oxides from driving cars in real-world driving conditions can be measured with very high temporal resolution. The combination of GPS data, and carefully constructed NOx measurement with a gas analyser that samples tailpipe emissions, allowed the authors to map some interesting features of NOx emissions as they occur in the real world. This clearly shows that NOx emissions are spatiotemporally heterogenous: much higher emissions are observed when cars accelerate after roadbumps, traffic lights, etc.

I found that the introduction of the paper lacked a bit of context. The authors should sketch who they think would be interested in these measurements, and why. Can they

be applied by national emission agencies for better emissions estimates, or are they useful for inspection authorities checking whether vehicles comply with EU-standards? Such information is missing from the paper. For illustration: the introduction now ends with wahet is "beyond the scope of this paper", but it would be better to formulate clearly what will be addressed in the paper. Also, I strongly support the suggestion by Referee #1 who calls for a link of the observations to Euro emission standards for good perspective.

Another concern I have is on the accuracy and the precision of the ultimate data provided (g/s and g/km emitted). The authors did not make any statement on the uncertainties in measuring the NOx concentrations at the tailpipe. This should be repaired, and I also think an uncertainty estimate should be provided for the emission strength estimates.

IF the authors wish to make the case that their method is superior to a conventional technique as PEMS, I think it would be good if they attempt to quantify how much better the detail is that they can detect now. Would that be relevant given the potential application of these data for e.g. vehicle compliance monitoring?

Specific issues P2, L20: dependant → dependent

P3, L11: Bajaj et al (2002) should be entirely between brackets.

P4, L22: "the deposition concentrations" is formulated a bit weird. With deposition, most atmospheric scientist understand the process of atmospheric constituents being deposited at the Earth's surface. Here it appears to describe the constituents emitted to the atmosphere. Please clarify to avoid confusion.

P12, caption Figure 3: please also include in the caption that this figure holds for a diesel car.
* * *

---

## Author Comment (AC1) · 9 May 2018

**Author response to referee comments Atmos. Meas. Tech. Discuss., amt-2017-305**

We would like to thank the two referees for their detailed comments and suggestions to improve the manuscript. We hope that the re-submitted manuscript addresses all concerns at this stage.

**Title:** HIGH SPATIOTEMPORAL RESOLUTION POLLUTANT MEASUREMENTS OF ON-BOARD VEHICLE EMISSIONS USING ULTRA-FAST RESPONSE GAS ANALYZERS

**Notes about the organisation of this document:**

The comments of the two referees (R1 and R3, respectively) have been addressed sequentially, with changes signified by page and line number related to the original submission.

In the case of similar or overlapping content in the comments (e.g. a general comment is satisfied by the resolution of a specific comment, or if one response/action addresses multiple comments), we give the full details of the change only at the location of the comment which required the most significant changes. All other related comments make reference to the item with the full response, identified by comment number. The full response may be earlier or later in the document than the comment in question.

Black text (hereafter):

Referee comment [with additional surrounding text for necessary context given in square brackets]

Blue text:

**Response:**Our comments to the reviewers and the public**Action:**What we have done to change this in the text of the article

From Referee 1:

**Comment:** I recommend publication after minor changes, although I feel that the paper could gain further scientific relevance with a more in-depth data analysis of the driving experiments.

Although the authors wish to "keep the focus on the technique and the instrumentation" (page 3, line 6), a better data analysis will lead to more interesting conclusions or recommendations (e.g. for reducing traffic emissions by changing driving styles or adapting urban infrastructure), and would give more direction to follow-up research.

**Response:** We'd like to thank the reviewer for this comment. A number of changes have been made throughout the manuscript (see below) in an attempt to

give more direction to follow-up research using similar techniques, where the focus would be on real world emissions and not the technique and instrumentation (e.g. a model and measurement sensitivity study of different routes incorporating differing levels of urban and rural infrastructure would be very informative to policy makers).

Action: P7: Section 4.2 – 'Further Work' added, replacing P7, L18 "Further improvements have already been made to the spatial accuracy of the positioning data. Real-time Kinematic (RTK) GPS has been successfully implemented allowing an accuracy of <1 cm to be achieved in favourable conditions, with an increased 15 Hz logging frequency. This has further increased the benefit of the spatial emissions positioning of this technique.

This paper discusses the specific technique required to take high spatiotemporal emissions data but consideration of the implications of such data is important. GPS plots of emissions hot spots could provide important information for local air quality councils and transport planning authorities. However due to the relative set-up times of such testing it is unlikely to be a useful tool for in-service compliance checks. If used correctly, emissions maps could provide recommendations on changes to improve local air quality. The most obvious change, from a purely scientific point of view, would be the removal of all speed bumps as these cause transients that contribute significantly to high NOx levels. Their placement close to schools or pedestrian areas reinforces this argument due to the direct impact on local air quality. Such a change would however have other political and safety implications far beyond the scope of this paper. Further, improvements through synchronisation of traffic lights in order to streamline traffic flows may be possible. A greater uptake in driverless or at least intelligent vehicles that could feedback to a centralised traffic control centre would aid this. Drivers could also shoulder some responsibility for emissions improvement so education in better driving techniques or even the inclusion of a driving quality metric into licencing tests could be beneficial. Although this study includes several tests on two differing vehicles, the data set is insufficient to allow the above hypothesis to be tested. Further work in repetition of specific road elements in different vehicles, and employing different driving styles, is needed."

**Comment:** In Section 2 (page 4, line 5) it is explained that the emissions from the gasoline car are sampled "pre-muffler but post 3-way catalyst", which is in conflict with the description in Section 2.2 (page 5, line 1-3) which state that NO is sampled both upstream and downstream of the catalyst. In the latter case: can the authors present some numbers about their measured efficiency of the catalyst? Is this as expected? How does it react under transient driving conditions?

**Response:** To clarify, for this vehicle, there are two sampling points; one before the catalyst, and one after the catalyst but before the muffler. In this study, the analyzer measured from both sample locations, however, only post-catalyst data has been published. The text has been appropriately changed.

On gasoline vehicles, catalyst efficiency is usually measured to be 100% at all steady state conditions once catalyst conversion temperature has been reached. Tailpipe

emissions breakthrough is measured where transient pedal input upsets catalyst conditioning (reductants), and NOx is no longer completely converted.

Action: P4 L5 Text changed "(...) fitted through the vehicle floor to two sampling points, one before the three-way catalyst and one post catalyst but premuffler. For engine calibration applications, having pre and post-catalyst sampling is essential to understanding aftertreatment operation and excellent time alignment. For air quality applications, analyzing post-catalyst data is more beneficial. The diesel data {...}"

**Comment:** Page 2, line 11: abbreviation RDE is used, while only explained at page 5 (line 19) for the first time.

**Response:** Agreed, this should be explained at first use.

Action: P2 L11 Text changed to define RDE at point of first use: "(...) of a Real World Driving Emissions (RDE) test (...)"

**Comment:** Page 4, line 13-15: maybe it should be remarked that the diesel vehicle is not equipped with a catalyst.

**Response:** Yes, this would provide extra context to the reasons for emissions breakthrough of this vehicle.

Action: P4 L15-6 Text added: "The vehicle was in good repair and in current use with no form of NOx aftertreatment (e.g. a catalytic convertor). For fleet context (...)"

**Comment:** Page 4, line 32-33: "(...) as it was anticipated that the NOx emissions would be relatively low (NOx being mainly a byproduct of diesel-powered internal combustion engines)". I presume that the authors mean NO2 instead of NOx (and that therefore there was no need to use an NO2 converter).

**Response:** Yes, this is a mistake and will be changed.

Action: P4 L32-3 Text changed, NOx to  $NO_2$ : "as it was anticipated that the  $NO_2$  emissions would be relatively low ( $NO_2$  being mainly a byproduct of diesel-powered internal combustion engines)"

**Comment:** Page 5, line 27: "Figure 3 shows (...) increasing from 20 mg/s baseline when stationary". From the figure I would estimate 2 instead of 20 mg/s.

**Response:** Agreed, text has been changed in the document.

Action: P5 L27 Text changed to "Figure 3 shows (...) increasing from 2 mg/s baseline when stationary"

**Comment:** Page 5, line 27-32: The authors are measuring both NO and NOx, so I would expect some words about their findings in the NO/NOx ratio. Is the NO2/NOx ratio approximately stable (also during the acceleration phases)? If so, is it relevant to keep measuring NO2 in future experiments?

**Response:** The mechanisms by which NO and NO2 are created are different. For a gasoline vehicle, very little NO2 is produced during combustion, therefore, measuring NO only is sufficient. The ratio of NO2:NOx in a diesel vehicle varies during transients up to approximately 30% dependent on air-fuel ratio, engine speed, load amongst other factors. It is therefore important to measure both NO and NOx because the addition of the NOx converter allows the most dangerous pollutant, NO2, to be calculated from the difference; interesting for air quality purposes as NO2 is individually legislated. Whilst NOx-only may be sufficient for atmospheric research; NO2 is often desirable for Air Quality research.

**Action: No publishable data at this time.**

**Comment:** Section 3.2, page 6, line 9-11: "Using much slower conventional PEMS equipment (...), this highly time-resolved event would be significantly delayed, and smoothed out over a longer period making its location difficult to place." I would also remark that the duration is so short (~2s) that its magnitude would be missed by PEMS, having a response time of about 1 second.

**Response:** Very true, this is a key advantage of fast-response equipment. Having a fast enough analyzer ensures that the full magnitude of each emissions peak can be fully resolved.

Action: P6 L10-11 Text changed to: "(...) this highly time-resolved event would be significantly delayed, and smoothed out over a longer period. The true magnitude of this event would be missed due to its short duration and its spatial location would be difficult to place.

**Comment:** Section 3.2, page 6, line 12-13: To put things into perspective, I would include a remark that the current Euro emission standards are 60 mg/km (petrol) and 80 mg/km (diesel).

**Response:** Agreed, this will provide extra meaning to the data, however, the current standards are not necessarily relevant due to the age of these vehicles, so have provided current and calibrated values for both.

Action: P6 L9 – Text added: "{...} emissions peak (for reference, the current emissions standards – Euro 6 at time of publication – are 60 mg/km NOX for Gasoline and 80 mg/km Diesel) {...}"

**Comment:** Section 3.3, page 6, line 19-20: From Figure 2b I learn that also the petrol car has been driven over these speed bumps. Why not include a comparison between the petrol and diesel vehicle emissions regarding speed bumps, as more or

less promised on page 4, line 26: "(...) used for comparison of diesel and gasoline vehicle emissions."?

**Response:** We agree that this is a good comparison, and one that we have made. However, we would rather avoid starting a Diesel vs Petrol debate since the paper is focussed on the measurement technique.

Action: P4 L25-6 Text also added based on previous reviewer comments: "The GPS data for the London drive was set to 1 Hz (due to an earlier version of acquisition software), and the drive duration was 2.5 hours for the 53 km of this route (shown on the left in Figure 1). An urban route around Cambridge UK passing near continuous air quality monitoring stations was also designed and used for comparison of diesel and gasoline vehicle emissions (shown on the right in Figure 1)."

We can also provide the referee with some example data directly:

**PETROL EURO4**

DIESEL EURO5

**Comment:** Page 6, line 30: "illustrating that acceleration (...) alone is not a suitable proxy for emissions.". Why not? If I look at Figure 6a, I see two different velocity gradients (roughly 2216.0-2217.5s and 2218.5-2220.0s), corresponding to different emission peaks. If the sampling rate of the speed permits, it would be insightful to overplot this graph with the vehicle's acceleration.

**Response:** It is true that in this case, where the vehicle is not fitted with any after treatment system (catalyst, SCR, etc) the acceleration is a good proxy for the ppm emissions (as seen in the first figure below), however this will fall down quickly for any modern vehicle with after treatment. This can be seen in the following plots, a Diesel fitted with SCR at steady state (and slight deceleration) can produce tailpipe

NOx (confidential data), and an example from the gasoline vehicle included this paper, fitted with a catalyst, where breakthrough occurs only when the catalyst is poorly conditioned (pre-catalyst in blue, post in orange).